# Regeneration of Articular Cartilage Using Membranes of Polyester Scaffolds in a Rabbit Model

**DOI:** 10.3390/pharmaceutics14051016

**Published:** 2022-05-08

**Authors:** Maciej Baranowski, Monika Wasyłeczko, Anna Kosowska, Andrzej Plichta, Sebastian Kowalczyk, Andrzej Chwojnowski, Wojciech Bielecki, Jarosław Czubak

**Affiliations:** 1Department of Orthopedics, Pediatric Orthopedics and Traumatology, Gruca Orthopaedic and Trauma Teaching Hospital, Centre of Postgraduate Medical Education, Konarskiego 13, 05-400 Otwock, Poland; czubakjarek@gmail.com; 2Nalecz Institute of Biocybernetics and Biomedical Engineering PAS, Księcia Trojdena 4, 02-109 Warsaw, Poland; mwasyleczko@ibib.waw.pl (M.W.); achwojnowski@ibib.waw.pl (A.C.); 3Department of Histology and Embryology, Medical University of Warsaw, Chałubińskiego 5, 02-004 Warsaw, Poland; akosowska@wum.edu.pl; 4Faculty of Chemistry, Warsaw University of Technology, 3 Noakowskiego Str., 00-664 Warsaw, Poland; andrzej.plichta@pw.edu.pl (A.P.); skowalczyk@ch.pw.edu.pl (S.K.); 5Department of Pathology and Veterinary Diagnostics, Faculty of Veterinary Medicine, Warsaw University of Live Sciences, Nowoursynowska 159c, 02-787 Warsaw, Poland; wojciech_bielecki@sggw.edu.pl

**Keywords:** scaffolds, regenerative medicine, cartilage tissue engineering, articular cartilage, poly(l-lactide-co-ε-caprolactone), rabbit, cartilage regeneration

## Abstract

One promising method for cartilage regeneration involves combining known methods, such as the microfracture technique with biomaterials, e.g., scaffolds (membranes). The most important feature of such implants is their appropriate rate of biodegradation, without the production of toxic metabolites. This study presents work on two different membranes made of polyester (L-lactide-co-ε-caprolactone-PLCA) named “PVP and “Z”. The difference between them was the use of different pore precursors—polyvinylpyrrolidone in the “PVP” scaffold and gelatin in the “Z” scaffold. These were implemented in the articular cartilage defects of rabbit knee joints (defects were created for the purpose of the study). After 8, 16, and 24 weeks of observation, and the subsequent termination of the animals, histopathology and gel permeation chromatography (GPC) examinations were performed. Statistical analysis proved that the membranes support the regeneration process. GPC testing proved that the biodegradation process is progressing exponentially, causing the membranes to degrade at the appropriate time. The surgical technique we used meets all the requirements without causing the membrane to migrate after implantation. The “PVP” membrane is better due to the fact that after 24 weeks of observation there was a statistical trend for higher histological ratings. It is also better because it is easier to implant due to its lower fragility then membrane “Z”. We conclude that the selected membranes seem to support the regeneration of articular cartilage in the rabbit model.

## 1. Introduction

The articular surfaces of the bones are covered with articular cartilage (AC), which is made of hyaline cartilage connective tissue. Hyaline cartilage prevents the abrasion of bones, is resistant to friction, and facilitates movement. So, it is necessary to enable proper movement [1,2]. The cells of cartilage, chondrocytes, produce an extracellular matrix (ECM), which is mainly made of collagen II and proteoglycans. Chondrocytes are located in small spaces in the ECM, called lacunae. These spaces do not allow their migration to damaged sites. Musculoskeletal ailments caused by cartilage damage are common and they are also more often recognized. Most cartilage damage occurs as a result of trauma, an unhealthy lifestyle, or various diseases, such as osteoporosis or autoimmune disorders. They can damage the cartilage, causing pain, stiffness, movement limitations, and even disability. This can be direct trauma to the cartilage—most often a sharp strike to a bone or repeated microtrauma, the overload causing gradual damage. Repeated damage leads to the formation of defects on the surface. AC lacks blood supply and the neural system, so it has limited regenerative capacity. Moreover, the regenerate process becomes less effective as the human body ages. Leaving defects untreated leads to the development of a degenerative disease [1,2,3,4,5,6,7,8,9].

Treatment techniques currently used include microfractures (MF), mosaicplasty, osteochondral allograft transplantation, and autologous chondrocyte implantation. Microfractures are now the gold standard of treatment. This technique consists in making the defect within the cartilage damage deep enough to cause the outflow of mesenchymal stem cells (MSC) from the bone marrow. However, due to the specific structure of cartilage (lack of blood vessels, layered structure), none of the abovementioned techniques achieves satisfactory results. The resulting regenerated tissue is often of poor quality: is fibrous-like cartilage, or does not have an appropriate layered structure; therefore, it has reduced mechanical resistance. Some techniques are complex and time-consuming, making them unavailable for many patients. Another problem is the size of the defect. The larger the defect, the more difficult it is to repair with known techniques. Currently, research is focused on combining the microfracture technique with the simultaneous use of scaffolds (3D membranes) to improve the regenerative capacity of cartilage [10,11,12,13,14,15,16,17,18,19,20,21,22,23,24,25]. A scaffold is a spatial structure that takes the form of membranes, hydrogels, or nonwovens. Such an implant can be colonized by autologous chondrocytes or mesenchymal stem cells (MSCs). It serves as a temporary matrix that provides a suitable environment for cells that guarantee success in the cartilage regeneration process [16,26,27,28,29]. Such scaffolds have already been successfully applied in patients [10,26].

The ideal biomaterial should enable the transport of nutrients to chondrocytes and allow the elimination of metabolites; it should also be completely biodegradable and biocompatible. The degradation products should be non-toxic, non-inflammatory, and mechanically neutral (with adequate softness, stiffness, and roughness). These materials should also be resistant to the conditions in the body, such as pH and body temperature for a certain period. Membranes should have an appropriate microstructure (porosity, pore size, pore shape) [28] and allow for the formation of functional gap junctions and interaction with other cells and the extracellular matrix. The asymmetry of the membrane structure is extremely important—one surface of the membrane should have numerous and large pores, while the opposite surface should have as few and as small pores as possible. This prevents the cells from escaping from the substrate. The size of the pore is properly defined in the purpose of the research, among other things, as is the kind of cells for which the membranes are intended. For example, working with chondrocytes requires smaller pores than working with stem cells for chondrogenesis or with osteoblasts. However, the most important parameters are: biodegradation time; non-toxic, soft metabolites; and a safe degradation process without inflammatory reactions [10,13,16,27,28,29,30,31,32,33,34,35,36,37,38,39].

Materials for an implant are mainly made of synthetic or natural polymers or a combination of both (hybrid materials) Currently, commercial scaffolds for cartilage regeneration are made primarily of collagen. Due to defects in the natural polymers, these scaffolds do not meet the appropriate requirements (their rapid hydrolysis, low mechanical stability) that lead to the regenerated tissue being non-valuable fibrocartilage rather than hyaline cartilage [13,16,26,27,28,37,40]. Unlike natural materials, polyesters provide good mechanical properties and can be used to produce a variety of scaffold shapes using many techniques. They are biocompatible with good mechanical strength that hydrolyze into harmless components that are metabolized in the body and are easily removed from the organism [27,37,41,42,43]. Currently, research is focused on synthetic polymers, such as polycaprolactone (PCL) and poly (L-lactic acid) (PLLA). Various proportions and combinations of these biodegradable polymers can be used to achieve the desirable surface, mechanical, and structural properties [41,44,45,46,47]. Synthetic polyesters, such as poly(L-lactide-ε-caprolactone) copolymers, are biocompatible and completely bioresorbable. Their degradation products are non-toxic to cells and the major route of the first stage of degradation is hydrolysis. The degradation pathway is through monomers that are elastic and they do not damage the articular surface. The second stage of degradation is the conversion of the monomer to carbon dioxide and water. When polylactide membranes are broken down, lactic acid is formed, which induces inflammation. On the one hand, however, the use of a lactic acid copolymer reduces the concentration of lactic acid in the decomposition products; on the other, it allows the glass transition temperature of the copolymer to be lowered and, ultimately, by disturbing the regularity of the PLA structure, it accelerates the decomposition process [48,49]. An important advantage of co-polyester PLCA structures is that they do not lose their strength in the aquatic environment, and their mechanical and biological durability is significantly greater than that of the collagen substrate. The fact that caprolactone, which is part of the copolymer, has been used for many years to coat absorbable surgical sutures, is an argument in favor of the selected membranes [50]. Individual polyesters differ in biodegradation time—the rule is that the longer the carbon chain, the longer the decomposition time. Due to this mechanism, we can roughly estimate the time it takes for the entire substrate to convert to water and carbon dioxide [51,52,53]. Such membranes can be used in medicine where scientists and doctors are still looking for new scaffolds for the treatment of articular cartilage. 

The literature describes many methods for obtaining synthetic and hybrid scaffolds for tissue engineering. One of the most frequently used membrane preparation techniques is wet phase inversion. The properly formed polymer solution is immersed in a coagulation bath containing a non-solvent of the polymer. Due to the solvent and non-solvent exchange, the phase inversion takes place that gives a membrane. In these techniques, membranes with different porosities and pore sizes can be obtained. Furthermore, in this method, the pore precursor can be added to a previously prepared polymer solution or during membrane formation. It can promote the formation of larger pore sizes and higher porosity. The pore precursors are eventually removed from the scaffold using a suitable solvent (porogen leaching) [54,55,56,57]. 

In this article, we report an animal model study using two scaffolds made of polyester (L-lactide-co-ε-caprolactone) (PLCA). The membranes were prepared by the wet phase inversion method. The difference between them was the use of different pore precursors. It has been hypothesized that the stem cells will colonize the membranes; after that, these cells will differentiate and form hyaline cartilage. The aim was to describe the effect of using the MF method with novel and promising scaffolds to regenerate hyaline cartilage in a rabbit model. The study was conducted in three different time frames using the microscopic examination. Microanalytic studies of origin membranes and extracted residue were carried out as well in order to find out the rate of the molar mass change of PLCA during in vivo therapy. We also evaluated the practical aspects of implantation and surgical techniques.

## 2. Materials and Methods

### 2.1. Materials

#### 2.1.1. Membranes

Scaffolds were obtained from the Institute of Biocybernetics and Biomedical Engineering (BBE) of Polish Academy of Sciences. Both scaffolds were made of PLCA by the wet inversion phase method. The difference was the use of various nonwovens as macroporous precursors. Membranes were obtained according to the method presented in previous work [25].

In the first case, the macropore precursor was a nonwoven fabric made of polyvinylpyrrolidone 1.3 MDa (PVP). We named it “PVP” (Figure 1). It was received as follows: the PLCA and Pluronic polymers with 4:1 were dissolved in dioxane with constant stirring to obtain 10 wt.% concentration. Next, a polymer mixture was poured onto the glass base and then the PVP nonwoven layer was laid. Then another portion of polymers was poured on the nonwoven and again a second slice of nonwoven and third layer of membrane forming solution were added. All layers were pressed and the air was removed using a Teflon roller. The received membrane was gelled in a bath with deionized water with ice (about 4 °C). The prepared membranes were stored in 70% ethanol. It is important to protect the PVP nonwoven from water to prevent its dissolving.

In the second case, the macropore precursor was a nonwoven fabric made of gelatin. We named it “Z” (Figure 2). It was obtained in a similar way to the previous membrane. The PLCA and PVP 10 kDa polymers with 4:1 ratio were dissolved in chloroform with constant stirring to obtain 10 wt.% concentration. The scaffold was made analogous to “PVP” preparation. The only difference was that the gelation bath contained cooled methanol at 4 °C and the obtained membrane was treated with warm water (50 °C) to removed gelatin nonwoven. Similarly to “PVP” membrane, scaffolds were stored in 70% ethanol. The gelatin nonwoven needs to be protected from water. 

The SEM micrographs of “PVP (Figure 1) and “Z” (Figure 2) scaffolds present an irregular structure with macropores and a three-dimensional network of interconnected macropores from 20 to even 500 μm in diameter. Both scaffolds have perforated skin layers that allow cells to enter them. Furthermore, the addition of extra pore precursors of polymers Pluronic (“PVP”) and PVP 10 kDa (“Z”) affects the microporous morphology of the membranes. It ensures access to oxygen, nutritious substances, or allows for the removal of metabolic products from the interior of scaffolds. The average thickness of both scaffolds is about 700–1200 µm. According to biomedical applications, the porosity and pore size of scaffolds are critical factors. In the previous study, the degradation rate and porosity of membranes (before and after hydrolysis) were measured [25]. To determine the degradation rate of scaffolds, PBS and Hank’s balanced salt solution (HBSS) fluids were used. The hydrolysis time was 5 weeks for “PVP” and 4 weeks for “Z” due to their faster destructions at 37 °C. The loss of weight was observed for both membranes in the range from 17 to 72 of weight percentage. We have also observed that the decrease in pH was not rapid—that should not have a negative effect on the body, such as the occurrence of inflammation. Results showed that both membranes were characterized by high porosity, about 95%. After degradation, this value increased, especially for “Z” in HBSS.

#### 2.1.2. Rabbits

We used 27 white New Zealand male rabbits, weighing about 3–4 kg, and aged about 4 months. Before the start of the study, the animals were habituated to the new environment and caretakers. During the experiment, the rabbits were weighed weekly. The animals were kept under standard environmental conditions: air humidity 55 ± 10%, temperature 21 °C ± 2 °C, 15 air changes per hour, circadian rhythm (light/ dark)12/12 h. All activities performed on animals were in accordance with the principles of occupational health and safety in the laboratory and good laboratory practice. The participants in the study were properly trained and have many years of experience in this field. We split the animals into three groups. In all the knee joints of all the animals, we performed grade IV defects on the articular surface, according to the Outerbridge scale [58]. In group I, defects were created with the simultaneous implantation of a membrane made of a copolymer (l-lactide-co-caprolactone) “PVP”. After implantation, we waited for 8 weeks (observing 7 defects), 16 weeks (7 defects), and 24 weeks (7 defects). Then the joint was removed and the regenerate was assessed. In group II, defects were created with the simultaneous implantation of a membrane “Z”. After implantation, we waited for 8 weeks (observing 7 defects), 16 weeks (7 defects), and 24 weeks (7 defects). In group III (control group), defects were created and no other action was taken. After surgery, we waited for 8 weeks (observing 4 defects), 16 weeks (4 defects), and 24 weeks (4 defects).

### 2.2. Methods

We conducted the study, taking into account the International Cartilage Repair Society (ICRS) recommended guidelines for histological endpoints for cartilage repair studies in animal models and clinical trials [59].

#### 2.2.1. Implantation

All surgical procedures were performed under aseptic conditions of the operating room at the Animal Breeding Laboratory of the Medical University of Warsaw. The surgical operations were performed according to the scheme (Figure 3):(a)We administered general anesthesia intramuscularly (xylazine and ketamine—dose calculated according to the animal’s body weight; ketamine—0.4 mg/kg, xylazine 0.5 mg/kg).(b)We shaved the area to be operated.(c)We placed the animal on the side opposite the one to be operated, with the limb in abduction. Skin decontamination with iodine solution. We covered the operating area with sterile drapes, which were attached to the skin.(d)We prepared the scaffold (membrane “PVP” or “Z”—depending on group). Under sterile conditions, we removed the membrane immersed in alcohol in transport packaging and placed the membrane in sterile petri dishes filled with a 0.9% NaCl solution. Rinsed the membranes twice at 10-min intervals with a 0.9% NaCl solution (to get rid of alcohol). For each animal, membranes were prepared in a separate petri dish. This point was omitted in the control group (where no membrane was used).(e)Skin incision on the lateral side at the level of the knee joint, then the incision of subcutaneous tissue and the joint capsule. Dislocation of the kneecap to the medial side, providing access to the articular surface of the femur. We created 2 defects located symmetrically at both condyles (each condyle had one defect) of the femur (load bearing area), using a chisel (Figure 3). Defect (0.1 cm^2^) of cartilage and bone to a depth of 3 mm. Confirming a full-depth defect penetrating the bone marrow by the occurrence of bleeding from the base of the defects.(f)We rinsed the defects with sterile 0.9% NaCl solution and dried them (to get rid of any debris). We cut a scaffold to the size of the defect and inserted a scaffold inside the defect so that the more porous surface of the membrane was in direct contact with the bone marrow.(g)After filling the defects with membranes, we applied the tissue glue (TisselLyo) to the surface of the defects according to the manufacturer’s instructions.(h)We waited for the adhesive to bond with the surrounding tissues. We moved back the kneecap into the correct position. We sutured the articular capsule with a Vicryl 3/0 suture, subcutaneous suture—Vicryl 4/0, skin suture—Ethilon 4/0 (to be removed in 14 days). We covered the postoperative wound with iodoform. Sterile dressing. We applied soft dressing from the ankle to the groin for 24 h.(i)Postoperative administration of antibiotics (Enrofloxacin) and analgesics (Metamizole) for 2 days.(j)Due to animal welfare reasons, we operated the second knees of the animals after the previously planned time (8 or 16 weeks, depending on the group).

#### 2.2.2. Aftercare

The aftercare was provided in an individual supervision room with electronically controlled temperature, humidity, and exposure time. To ensure maximum comfort and safety for the animals and to minimize the possibility of mistakes, each rabbit had its own separate cage. The animals could move freely around the cage throughout the observation period. Each rabbit had a daily observation card with vital parameters (general and local condition, temperature, body weight), date of surgery, and date of planned termination.

#### 2.2.3. Termination

After the set time of 8, 16, 24 weeks of observation, the rabbits were terminated. Termination procedures were performed according to the scheme (Figure 4):(a)We administered general anesthesia intramuscularly (dose calculated according to the animal’s body weight—ketamine—0.4 mg/kg, xylazine 0.5 mg/kg).(b)Euthanasia by intravenous administration of Morbital.(c)We shaved the knee for surgery; disinfection of shaved areas.(d)Access to the knee joint (skin and deeper tissues were cut, as in the case of surgery). Extraction of the distal femur. Marked the place of defects/membrane insertion with ink. Cut the condyles from the rest of the bone and placed in a sterile transport container for further examination. Lateral condyle was taken to the histopathology examination, medial condyle to GPC.

#### 2.2.4. Gel Permeation Chromatography (GPC)

GPC is the analytical technique that allows to detect and characterize qualitatively polymer chains soluble in eluent used. It provides number and weight average molar masses and the dispersity index of polymers. The chains of various lengths are separated on the gel column with respect to their hydrodynamic radii and the time of elution is correlated with molar masses at the peak of narrow polystyrene standards. The measurement was carried out with the system by Viscotek composed of GPC max and TDA 305, equipped with Jordi Lab DVB column (mixed bed) and refractometer. Dichloromethane was used as eluent with a flow rate of 1 mL/min at 30 °C. Sample concentration was in the range of 2–4 mg/mL and injection volume was 50 or 150 µL.

Preparation of the material for testing was carried out according to the following scheme:(a)Scraping and cutting only cavities from previously taken specimens;(b)Flushing in a physiological saline solution—flushing out physiological fluids;(c)Rinsing in hexane to extract fats;(d)Washing in CDCl3 (deuterated chloroform)—deproteinization;(e)Rinsing in methylene chloride—extraction of membranes (membrane residues);(f)Filter through a syringe filter with a PTFE membrane, with a porosity of 0.2 μm to get rid of bits of remaining cartilage (insoluble parts).

The GPC test was performed for 5 samples with “PVP” and 5 samples “Z” membrane 8 weeks after implantation; 3 samples with “PVP” and 3 samples “Z” membrane 16 weeks after implantation.

#### 2.2.5. Histopathology

The study was carried out according to the following scheme:(a)Collected condyles were immersed in 10% formalin.(b)Descaling—while waiting for the descaling, checks were made on the degree of descaling by trying to puncture the tissue with a needle every few weeks. After decalcification, the damaged areas were more visible.(c)The decalcified sections were dehydrated and embedded in paraffin (Paraplast sigma).(d)The material was cut into pieces with a thickness of 4 μm.(e)Paraffin sections were stained by the routine hematoxylin–eosin method.(f)The regenerates were evaluated (ICRS microscopic scoring system) under a light microscope by two trained and blinded observers. Each observer rated 1 regenerate 3 times at weekly intervals. The final score is the average of 3 measurements [60].

#### 2.2.6. Statistics

R software with vegan [61], coin [62], and FSA [63] packages was used to answer the research questions. Kendall’s *W* coefficient of concordance, Kruskal–Wallis tests with post-hoc Dunn’s tests, and Holm’s correction for multiple comparisons were performed. Given the small sample size in the study, Kruskal–Wallis *p* values were approximately estimated using 10,000 Monte-Carlo simulations, with bootstrap resampling [64]. Similarly, for Kendall’s *W*, 10,000 permutations tests were carried out to estimate null distribution of χ^2^ [65]. The significance level was set as α = 0.05, while *p* values between 0.05 and 0.1 were treated as tendencies of significance.

## 3. Results

### 3.1. Destruction of PLCA Materials in Membranes Analyzed by Means of GPC

The weight average (M_w_) and number average (M_n_) of molar masses of the initial PCLA copolymer introduced to the membranes were 138,200 g/mol and 78,100 g/mol, respectively. We examined the copolymer material extracted after 8 and 16 weeks of implantation from 16 rabbit joints in total. The average molar masses were measured by means of GPC and calculated with respect to PS standards for all samples. Each sample contained several populations of macromolecules present at fractions. Since the molar mass distributions partially overlapped to facilitate statistical analysis, we decided to calculate the mean values for M_w_ and M_n_ for whole multimodal macromolecular distribution of all materials from each sample separately (Table 1). Exemplary, selected molar mass distributions are shown for illustration in Figure 5. Finally, we calculated the mean values of M_w_ for each type of membrane and implantation period. The data with standard deviation bars are presented in Figure 6.

We performed the GPC examination on sections collected only from cavities to prove that the membranes stayed at the primary site of implantation. Because in every sample we detected fractions of polymers, we concluded that membranes remained in the cavities. This confirms that the surgical technique developed by us and used in the study meets the requirements.

### 3.2. Histopathology

#### 3.2.1. Differences in Membranes Performance

Kruskal–Wallis tests were performed, for ratings provided by each histologist, in each time-point (at 8, 16, or 24 weeks), to examine the differences in recovery between the three groups: a group treated with the membrane PVP (*n* = 7; the PVP group), a group treated with the membrane Z (*n* = 7; the Z group), and the control group (*n* = 4).

#### 3.2.2. Histologist 1

There were no significant differences between studied groups at the 8-week mark—*H*(1) = 2.35, *p* = 0.310, ε^2^ = 0.14. However, there was a trend toward a significant difference of ratings at the 16-week mark—*H*(1) = 5.53, *p* = 0.057, ε^2^ = 0.33, and a significant difference at the 24-week mark—*H*(1) = 5.86, *p* = 0.045, ε^2^= 0.34. Post-hoc tests at 16 weeks showed a trend for better ratings in the Z group than in the control group—*Z* = 2.35, *p* = 0.056. There were no differences between the PVP group and the control group—*Z* = 1.42, *p* = 0.309, and no differences between the PVP group and the Z group—*Z* = 1.08, *p* = 0.278. At the 24-week mark, there was a trend for higher ratings in the PVP group than in the control group—*Z* = 1.97, *p* = 0.099. No differences between the Z group and the control group—*Z* = 0.16, *p* = 0.872, or between the Z group and the PVP group, *Z* = 2.12, *p* = 0.103, were observed (Figure 7).

#### 3.2.3. Histologist 2

For histologist 2, there were no differences at the 8-week mark—*H*(1) = 1.96, *p* = 0.150, ε^2^ = 0.23 or the 16-week mark—*H*(1) = 1.04, *p* = 0.623, ε^2^ = 0.06. However, the Kruskal–Wallis test was significant at the 24-week mark—*H*(1) = 9.46, *p* = 0.004, ε^2^ = 0.56. Post-hoc tests for the 24-week mark revealed that histologist 2 rated the healing stage of the membrane PVP—*Z* = 2.48, *p* = 0.027, and the membrane Z—*Z* = 2.98, *p* = 0.009, as superior to the healing properties without a membrane. At the same time there were no differences between the healing properties of the membrane PVP and the membrane Z—*Z* = 0.59, *p* = 0.552 (Figure 8).

It should be noted that the histologic images improve with time when both membranes are used (Figure 9). This proves that the tested scaffolds support cartilage regeneration.

## 4. Discussion

In our work, we examined scaffolds that have potential use for cartilage tissue engineering. Our scaffolds are characterized by biocompatibility, degradability, and adequate structure. They have perforated top layers with pore diameters of more than 20 μm that allow MSCs to penetrate inside the membrane. The interiors of both membranes (Figure 1 and Figure 2) shows a network of interconnected macropores with an appropriate pore diameter larger than 300 µm, necessary for the chondrogenesis of MSCs [16,66,67,68]. Their structures provide an appropriate environment for the proliferation, migration, and adhesion of cells. The semipermeable structure assures nutritious, oxygen transport, and metabolic products. Furthermore, the bottom skin layers of the scaffolds are dense, preventing cells from getting out from scaffolds [28].

During the implantation phase, we noticed that membrane “Z” was more fragile and disintegrated easier compared to membrane “PVP”. Because of this, it was easier to place and hold the “PVP” scaffold at the chosen localization (defect). This feature can be of great importance during implantation, not with glue but with threads, e.g., in the human knee.

We created (according to the International Cartilage Repair Society (ICRS) recommended guidelines for histological endpoints for cartilage repair studies in animal models and clinical trials) relatively large (0.1 cm^2^) and deep (3 mm) defects (Figure 3), which, especially after the first observation period (8 weeks), could have resulted in the lack of statistical significance of the cartilage regeneration assessments between groups. However, the extended observation time showed that the scores of regenerates without the use of a membrane were worse with each subsequent observation period and the scores of regenerates with membranes improved over time (Figure 7 and Figure 8). We believe that this is because the cartilage in the control group is mechanically weak, despite the fact that, in the histological picture, it was initially similar to the other groups. After some time, it degenerated and had a worse end result. We suspect the reason for that could be the stem cells in the control group having settled only superficially around the defect. Due to all of the above, we believe that the observation times in the rabbit model should be extended to at least 24 or even 36 weeks.

For rabbit welfare reasons, only one knee joint was operated on under each anesthesia. This was followed by a scheduled observation period for the group, during which the animal regained full strength. Only after the observation period was the other knee operated on. Therefore, different membranes were loaded differently during the various observation periods. The movements of the rabbits were not the same after surgery on one limb and again were different after surgery on both limbs. There were also individual characteristics in pain perception and soft tissue recovery and body weight that affected joint loading.

We suspect that PLCA copolymer degrades in the body mostly through the hydrolytic degradation [69]. The resulting lactic and caproic acids, which are naturally occurring products in the body, are subsequently metabolized into CO_2_ and H_2_O and eliminated from the healthy body. After implantation, the hydrolysis and degradation of polymer starts. In this study, the two stages of degradation were seen after 8 and 16 weeks. In both periods, residual PLCA materials were characterized by means of GPC in order to estimate average values of their molar masses. There was a decrease of mean M_w_ of both membranes during time (Figure 6).

Different populations of molecules arose in given samples (Figure 5). We do not know why this happened. We assume this was due to limitations of in vivo testing. The decomposition of a polymer molecule begins at its edge and continues deeper into it. What is more, polymers break down from smallest molecules (dispersion). The number of edges may be influenced by the implantation technique (too tight or loose filling of the defect in different individuals). Additionally, the presence of different amount of tissue glue, which initially isolates the membrane from the intra-articular environment, may have an influence. Each rabbit had a different enzymatic activity, each moved differently after the operation of membrane implantation, and each stressed the limb differently. There was also different body weights of animals and individual regenerative properties. Hence, there may have been some irregularities in the breakup. The small percentages of molecules with higher molecular weights than the original were likely contaminants that could not be removed during the preparation of the chromatography samples. Despite so many different variables that were difficult to isolate and study separately, a statistical regularity could be observed with exponential progress in degradation (Figure 6) [70].

One of the PVP 8 samples differed from the others because it only contained two major populations of molecules and a small mass. During the observation of this individual, we did not notice any of its properties that could affect such a result. We believe that he was either individually overactive in enzymes or that we made a laboratory error in preparing the sample (Table 1).

The GPC test carried out each time detected the residual membrane, which shows that, in none of the tested cases, the membrane migrated out of the defect and, thus, the implantation technique was appropriate (Table 1). 

In our study, histopathological evaluations varied among histologists (Figure 7 and Figure 8). It follows that the histopathological assessment may not be reproducible. It should be considered whether the ICRS scale requires modifications or corrections. We plan to subject the histopathological samples to further histopathological examinations using other scales. The question remains open as to whether, instead of histopathological tests, a better indicator of cartilage regeneration would be to test the level of glycosaminoglycans or collagen II concentration. Moreover, magnetic resonance imaging (MRI) should be considered when assessing articular cartilage. MRI is relatively non-invasive and can be performed at multiple time points in the same animal, thus enabling long-term follow-up assessments. MRI can show tissue overgrowth and bone edema, which are common complications of cell therapy procedures. Current imaging techniques may indirectly suggest hyaline cartilage formation, but these images are not always directly related to the histological findings [71,72].

The animal model is different from the clinical situation in humans. Humans are treated sometime after the defect develops, whereas, in our study, healthy animal joints were treated immediately after the defect developed. Cartilage regeneration was also affected by the fact that the animals fully loaded the limb immediately after surgery, which is the opposite of how humans recover. Hence, the evaluation of regeneration in humans in the future may be different.

Our work was carried out in accordance with the 3 R-rule (reduction, replacement, refinement). For this reason, we created two defects in one knee joint—one in the medial condyle and one in the lateral condyle, symmetrically (Figure 3). In groups I and II, in each joint, the regenerated defect from the lateral condyle was subjected to a microanalytical examination, the regenerated defect from the medial condyle was subjected to a histopathological examination (Figure 4). In the control group, both defects in each joint were subjected to histopathological examination. The defects in the control group in the same joints differed in terms of assessment, as did the defects in the two different joints. This means that even though the defects were located in the same joint, the regeneration process was different. For this reason, we treated these regenerates as separate statistically accountable values. We believe that since the process of regeneration of two defects in one joint differs, this process is influenced not only by the individual characteristics, the site of the defect (one defect on one condyle symmetrically), and its size (the same in each joint), but also by means that have not yet been recognized. The mechanics themselves and the way rabbits load the joints probably play a major role. 

According to the International Cartilage Repair Society (ICRS) recommended guidelines for histological endpoints for cartilage repair studies in animal models and clinical trials, we were looking for any implanted (foreign) material. During preparation of the samples, we did not find any pieces of membranes; moreover, we found no evident residues of previously implanted material in any of the histopathological samples. Based on such results and the GPC examination (Table 1), we assume that this is because the entire polyester has been degraded into the form of very short polymer chains that will be easily and completely degraded in a short time period into the products that can be resorbed by the body.

## 5. Conclusions

In this work, two types of membranes were tested to demonstrate their effectiveness in supporting the regeneration of articular cartilage in rabbits. The results of microanalytical and histological examinations showed that both scaffolds can support cartilage regeneration. The biodegradation process of the studied membranes is progressing exponentially, causing the membranes to degrade at the appropriate time. The proposed implantation technique is fully sufficient to properly place the scaffolds in the place chosen by the operator. The “PVP” membrane is better due to the fact that after 24 weeks of observation there was a statistical trend for higher histological ratings. It is also better because it is easier to implant due to its lower fragility then membrane “Z”. We can conclude that the selected membranes seem to support the regeneration of articular cartilage in the rabbit model.

## Figures and Tables

**Figure 1 pharmaceutics-14-01016-f001:**
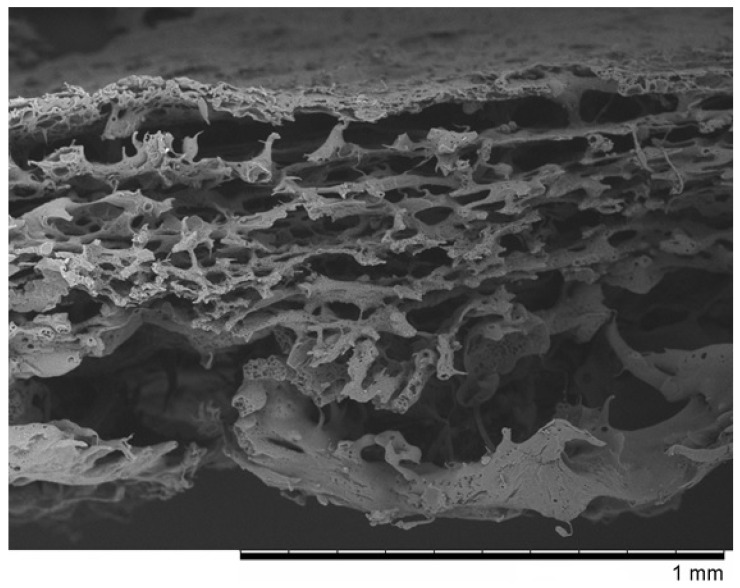
The SEM photomicrographs of the “PVP” membrane.

**Figure 2 pharmaceutics-14-01016-f002:**
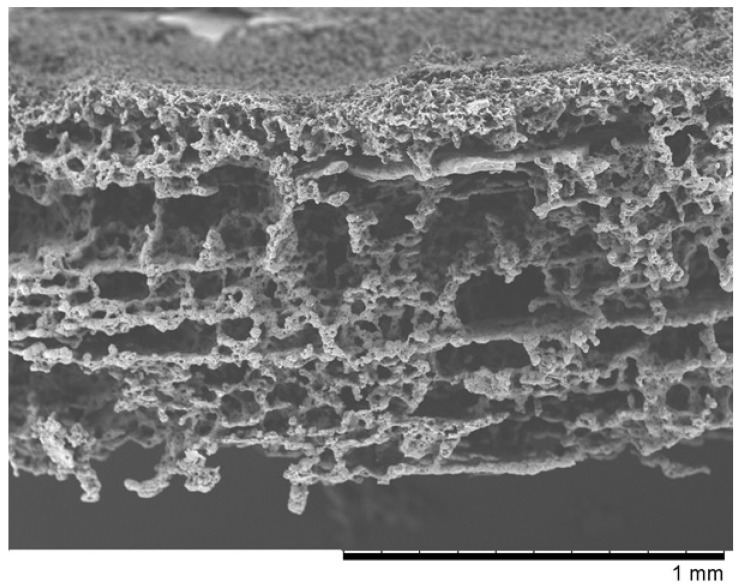
The SEM photomicrographs of the “Z” membrane.

**Figure 3 pharmaceutics-14-01016-f003:**
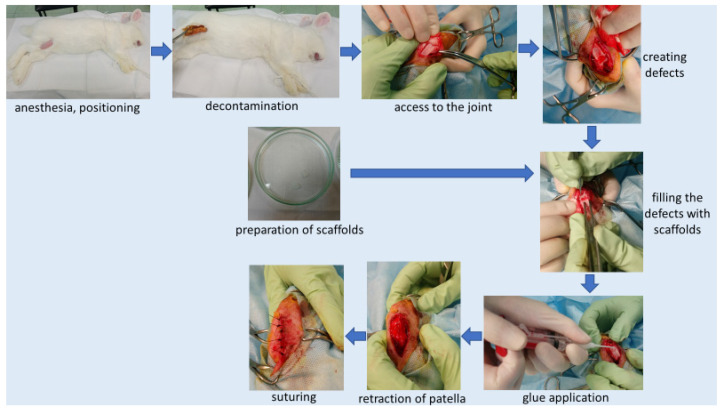
Implantation scheme.

**Figure 4 pharmaceutics-14-01016-f004:**
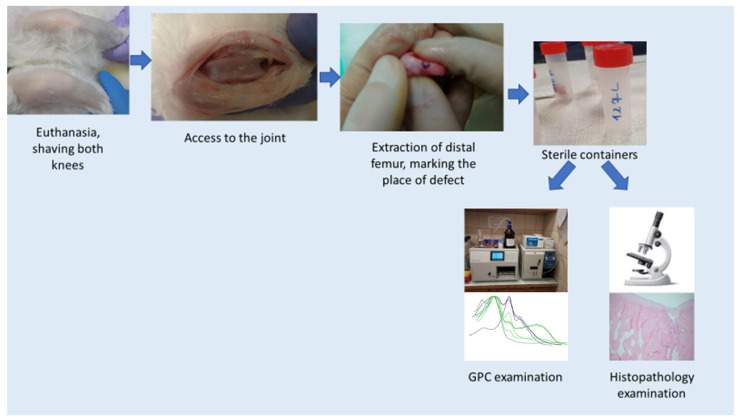
Termination scheme.

**Figure 5 pharmaceutics-14-01016-f005:**
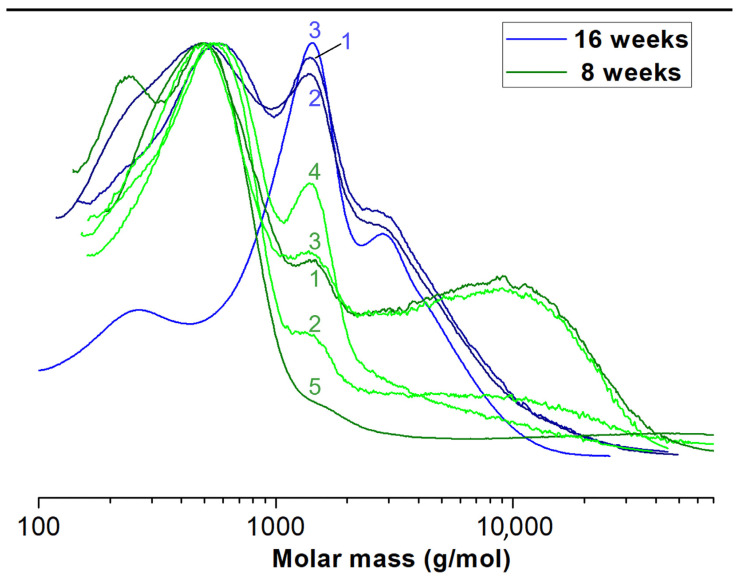
Exemplary molar mass distributions of extracted “PVP” membrane residue.

**Figure 6 pharmaceutics-14-01016-f006:**
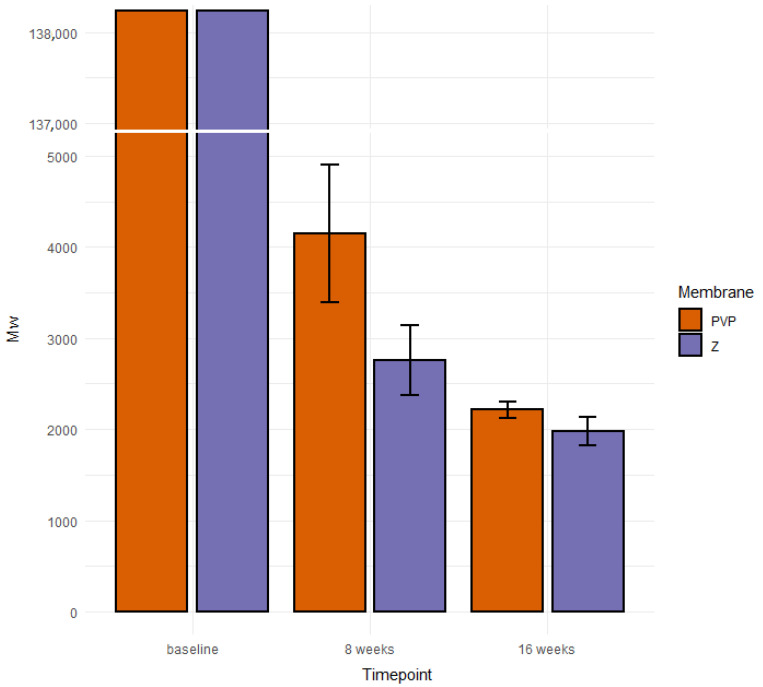
Average values of M_w_ of PLCA contained in origin membranes and residue materials.

**Figure 7 pharmaceutics-14-01016-f007:**
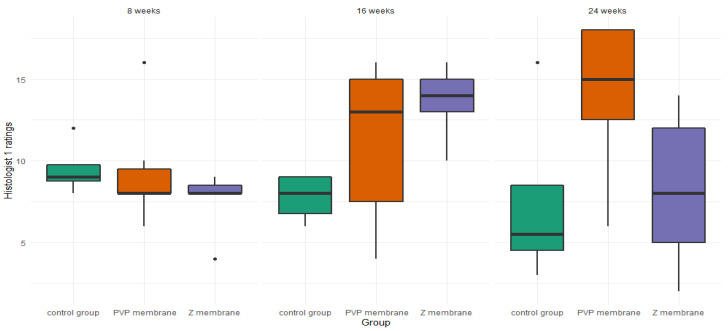
Boxplot of histologist 1 ratings, in the three studied groups, at three time points.

**Figure 8 pharmaceutics-14-01016-f008:**
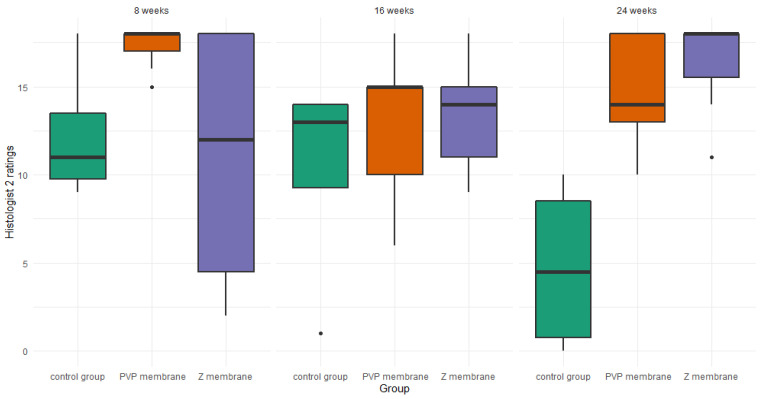
Boxplot of histologist 2 ratings, in the three studied groups, at three time points.

**Figure 9 pharmaceutics-14-01016-f009:**
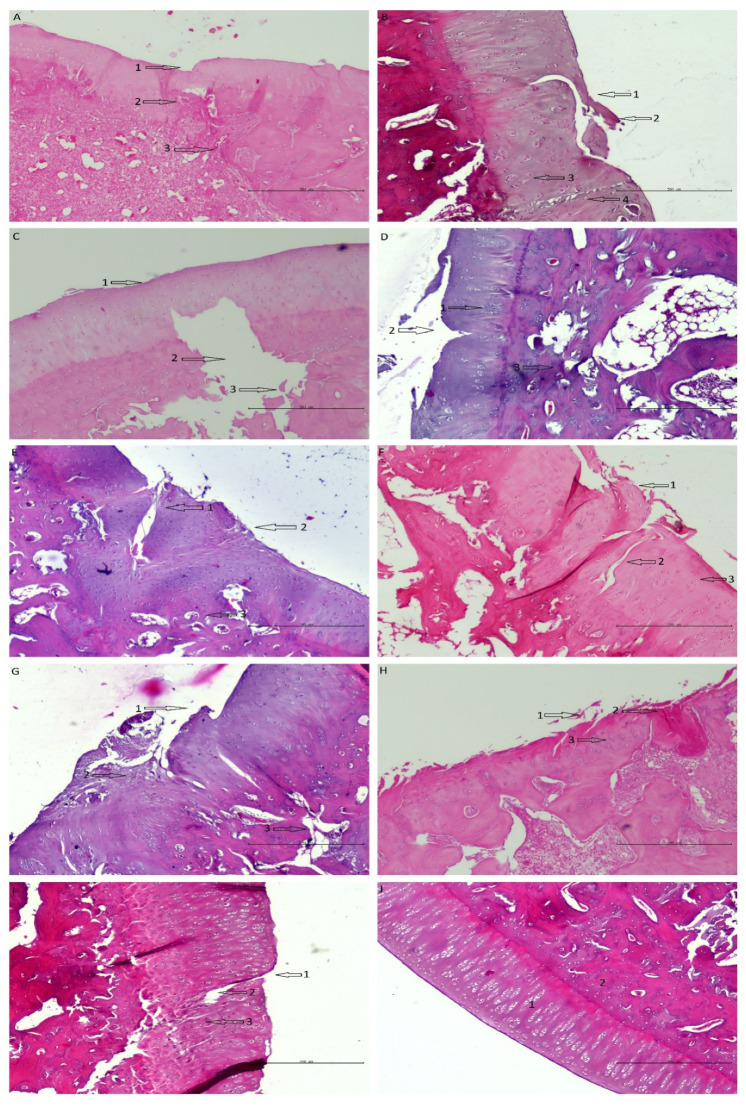
Shows histological samples of the cartilage of the rabbit of the studied groups. The following observations were made by one histologist. (**A**) PVP group after 8 weeks of observation—the defect is easily visible; (1) the surface has irregularities, (2) cells are disorganized, (3) increased remodeling of subchondral bone. (**B**) Z group after 8 weeks of observation—the defect is easily visible; (1) the surface has irregularities, (2) cartilage necrosis, (3) cells are distributed in clusters (4) chondral fracture reaching the subchondral bone. (**C**) Control group after 8 weeks of observation—the defect is easily visible; (1) the surface has irregularities, (2) massive loss of subchondral bone, (3) Bone fractures with the separation of necrotic fragments. (**D**) PVP group after 16 weeks of observation; (1) cells are distributed in clusters, (2) cartilage fracture, the surface has irregularities; (3) porous subchondral bone in some places, otherwise normal. (**E**) Z group after 16 weeks of observation—the defect is easily visible; (1) cells are distributed irregularly, (2) the surface is torn, (3) porous subchondral bone in some places, otherwise normal. (**F**) control group after 16 weeks of observation—the defect is easily visible; (1) the surface is torn, (2) chondral fractures reaching the subchondral bone, (3) cells are distributed irregularly. (**G**) PVP group after 24 weeks of observation; (1) the surface is torn/has irregularities, (2) cartilage fractures, cells are distributed irregularly, (3) increased remodeling of subchondral bone. (**H**) Z group after 24 weeks of observation; (1) the surface is torn, (2) necrosis, (3) cells are distributed in columns. (**I**) Control group after 24 weeks of observation; (1) the surface has irregularities; (2) full thickness chondral fracture; (3) cells are distributed irregularly. (**J**) Histological image of correct hyaline cartilage in rabbit model—(1) cartilage, (2) subchondral bone.

**Table 1 pharmaceutics-14-01016-t001:** Results of the GPC examination.

Kind of Membrane	Time of Implantation (Weeks)	Sample Nr	Mean Mw
PVP	8	1	5155
PVP	8	2	4816
PVP	8	3	4737
PVP	8	4	1896
PVP ^1^	8	5	24,238
PVP	16	1	2387
PVP	16	2	2176
PVP	16	3	2088
Z	8	1	3678
Z	8	2	3167
Z	8	3	3022
Z	8	4	2490
Z	8	5	1432
Z	16	1	2139
Z	16	2	2127
Z	16	3	1663

^1^ We reject this sample for statistical reasons—the result of the mean Mw clearly deviates from the other samples.

## Data Availability

Not applicable.

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
