# Peer review of "Regeneration of Articular Cartilage Using Membranes of Polyester Scaffolds in a Rabbit Model"

_pharmaceutics, 2022, doi:10.3390/pharmaceutics14051016_

Round 1

Reviewer 1 Report

The revision is well edited according to reviewers’ comments. A specific comment is noted below.

(Abstract) Please reduce background information. Instead add conclusion and/or implication at the end. Also, please shortly mention the difference between “PVP” and “Z”.

Author Response

Response to Reviewer 1 Comments

Point 1:
(Abstract) Please reduce background information. Instead add conclusion and/or implication at the end. Also, please shortly mention the difference between “PVP” and “Z”.

Response 1:

Thank you for your detailed revision of our manuscript. We reduced background information, added conclusion and the difference between 2 membranes.

New Abstract

"Currently, one of the promising methods for cartilage regeneration is to combine known methods, such as the microfracture technique with biomaterials like scaffolds (membranes). The most important feature of such implants is their appropriate rate of biodegradation, without the production of toxic metabolites. This study presents work on two different membranes made of polyester (L-lactide-co-ε-caprolactone - PLCA) named “PVP and “Z”. The difference between them was the use of different pore precursors - polyvinylpyrrolidone in “PVP” scaffold and gelatin in “Z” scaffold. These were implemented in the articular cartilage defects of rabbit's knee joints (defects were created for the purpose of the study). After 8, 16 and 24 weeks of observation, and the subsequent termination of the animals, histopathology and gel permeation chromatography (GPC) examinations were performed. Statistical analysis proved that the membranes support the regeneration process. GPC test proved that biodegradation process is progressing exponentially, causing the membranes to degrade of appropriate time. The surgical technique we used meets all the requirements without causing the membrane to migrate after implantation. “PVP” membrane is better due to the fact that after 24 weeks of observation there was a statistical trend for higher histological ratings. It is also better because it is easier to implant due to its lower fragility then membrane “Z”. We conclude that the selected membranes seem to support the regeneration of articular cartilage in the rabbit model."

Reviewer 2 Report

Thanks for answering my comments 

The article is OK for publication. The experiment is described clearly, which was one of my major concerns.

Author Response

Once again thank you for your detailed revision of our manuscript. 

All changes have significantly improved the manuscript.

This manuscript is a resubmission of an earlier submission. The following is a list of the peer review reports and author responses from that submission.

Round 1

Reviewer 1 Report

The manuscript is revised satisfactorily. However, it is important to clearly mention page no. and if possible line no. where the changes have been made while answering the comments. 

Reviewer 2 Report

Thank you for the interesting paper. This is quite interesting but has a major flaw that needs to be addressed before publication. 

-There are many instances of abrupt finishing of sentences like"Defects that 
were created for the purpose of the study." in the abstract. please go through the article and fix it.

-the abstract should contain your conclusion.

-in the literature review, please be accurate. For example, "chondrocytes, produce an extracellular matrix (ECM), which is mainly made of collagen and proteoglycans" has to be changed to collagen type II.

-Please mention the reference for this sentence "The resulting regenerated tissue is fibrous cartilage"

-you don't need to explain to the reader what biocompatibility is. please remove " (well tolerated by the human body)"

-what's e MF method in line 124?

-please explain the "wet inversion phase method" briefly in your introduction section 

-Why PLCA was used in the first place?

-perhaps the biggest flaw in this paper is the lack of different histological staining. Please clarify.

  • I really don't understand why you have no staining of anti-collagen ii on your histological slides? Can you explain why? don't you want to make hyaline cartilage?
  • -how can you conclude "In our work, we use scaffolds that meet adequate parameters for cartilage tissue engineering."? you are not showing enough results that there is cartilage there!
  •  

Reviewer 3 Report

This is an animal study of articular cartilage regeneration using 2 kinds of polyester membranes. In histologic examination, PVP and Z showed positive results compared to control. However, cell distribution and structural morphology were not satisfied compared to intact tissue. Authors discussed the potential problem and alternative methods. Specific comments and questions are noted below.

(Abstract) Please mention more specific results in the abstract.

(Line 77-85) Please supply references to support the importance of asymmetric membrane. 

(Introduction) What is your hypothesis? 

(Line 124) What is MF method?

(Line 159-160) Is the addition of Pluronic named "PVP" not "Z"? For "PVP", 1.3 MDa PVP was used. Please clarify your statement. 

(Section 2.1.1) The reason why you prepared 2 kinds of PLCA membranes is not clear. As you mentioned in the introduction, does they have a different pore structure (symmetric vs. asymmetric)? Only difference is hydrolysis time (4 weeks vs. 5 weeks). What is the advantage to use each?

(Section 2.1.2) Animal number is not matched with the number of your experimental groups.

(Section 2.2.1) Please describe more detail information of grade IV defects in terms of the size of membrane, the location of defects (load-or non-load-bearing area), and the number of defects (one knee or both knees).

(Line 291) Please describe how to analyze ICRS microscopy scoring system in terms of the number of observers, how many times to score, and blinded or non-blinded.

(Section 3.1) I suggest adding at least one more histologist and calculate overall result of histology. Also, Safranin-O staining is needed to add to evaluate proteoglycan integrity.

(Figure 9) Please add the control group and high magnitude images to see cellularity (migrated cells). Please add a scale bar too.